# Data Distributional Properties Drive Emergent In-Context Learning in Transformers

**Stephanie C.Y. Chan**
DeepMind

**Adam Santoro**
DeepMind

**Andrew K. Lampinen**
DeepMind

**Jane X. Wang**
DeepMind

**Aaditya K. Singh**
University College London

**Pierre H. Richemond**
DeepMind

**James L. McClelland**
DeepMind, Stanford University

**Felix Hill**
DeepMind

## Abstract

Large transformer-based models are able to perform in-context few-shot learning, without being explicitly trained for it. This observation raises the question: what aspects of the training regime lead to this emergent behavior? Here, we show that this behavior is driven by the distributions of the training data itself. In-context learning emerges when the training data exhibits particular distributional properties such as burstiness (items appear in clusters rather than being uniformly distributed over time) and having large numbers of rarely occurring classes. In-context learning also emerges more strongly when item meanings or interpretations are dynamic rather than fixed. These properties are exemplified by natural language, but are also inherent to naturalistic data in a wide range of other domains. They also depart significantly from the uniform, i.i.d. training distributions typically used for standard supervised learning. In our initial experiments, we found that in-context learning traded off against more conventional weight-based learning, and models were unable to achieve both simultaneously. However, our later experiments uncovered that the two modes of learning could co-exist in a single model when it was trained on data following a skewed Zipfian distribution – another common property of naturalistic data, including language. In further experiments, we found that naturalistic data distributions were only able to elicit in-context learning in transformers, and not in recurrent models. In sum, our findings indicate how the transformer architecture works together with particular properties of the training data to drive the intriguing emergent in-context learning behaviour of large language models, and how future work might encourage both in-context and in-weights learning in domains beyond language.[1]

## 1 Introduction

Large transformer-based language models show an intriguing ability to perform **in-context learning** (Brown et al., 2020). This is the ability to generalize rapidly from a few examples of a new concept on which they have not been previously trained, without gradient updates to the model. In-context learning is a special case of few-shot learning in which the output is conditioned on examples from a 'context', and where there are no gradient updates. It contrasts with **'in-weights' learning**, which is

---

[1]Code is available at: https://github.com/deepmind/emergent_in_context_learning

36th Conference on Neural Information Processing Systems (NeurIPS 2022).

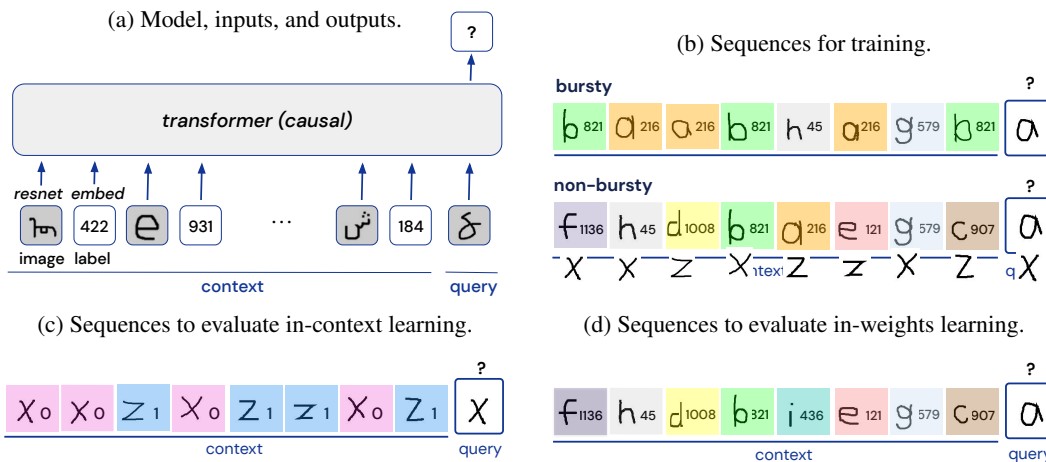

Figure 1: Experimental design, as described in Section 2. (a) For each experiment, a transformer model is trained on sequences of image-label pairs. The model is trained to minimize the loss on predicting the label corresponding to the final 'query' image. (b) In training, image-label mappings are fixed across sequences, in contrast to few-shot meta-training. The training data consist of a mix of 'bursty' and 'non-bursty' sequences. Bursty sequences, featuring multiple occurrences of the same classes, can be solved by learning labels across sequences (in-weights learning), or referring back to the context (in-context learning). Non-bursty sequences were composed of i.i.d. images. (c) To evaluate few-shot in-context learning, the model is presented with a standard few-shot sequence. The holdout image classes were never encountered in training, and are randomly assigned to labels {0,1}. Thus the model must use the context to predict the query label. (d) To evaluate in-weights learning, the model is presented with sequences where the labels are the same as in training. However, the query class does not appear in the context. Thus, the model must used information stored in weights to predict the query label. In the example sequences, we add colors and use only Latin characters for visualization purposes.

the standard setting for supervised learning – this is slow (requiring many examples), and depends on gradient updates to the weights. Earlier work in the context of 'meta-learning' showed how neural networks can perform few-shot learning without the need for weight updates (Santoro et al., 2016; Vinyals et al., 2016; Wang et al., 2016). To achieve this, the researchers explicitly designed the training regime to incentivize in-context learning, a process sometimes called 'meta-training'. In the case of transformer language models, however, the capacity for in-context learning is *emergent*. Neither the model's transformer architecture nor its learning objective are explicitly designed with in-context learning in mind.

Here, we consider the question of how transformer language models are able to acquire this impressive ability, without it being explicitly targeted by the training setup or learning objective. The emergence of in-context learning in language models was observed as recurrent models were supplanted by transformers, e.g. in GPT3. Was the novel architecture the critical factor behind this emergence? In this work we explore this possibility, as well as a second: that a capacity for in-context learning depends on the *distributional qualities of the training data*.

This hypothesis was inspired by the observation that many natural data sources – including natural language – differ from typical supervised datasets due to a few notable features. For example, natural data is temporally 'bursty''. That is, a given entity (word, person, object, etc) may have a distribution that is not uniform across time, instead tending to appear in clusters (Altmann et al., 2009; Alvarez-Lacalle et al., 2006; Lambiotte et al., 2013; Neuts, 2007; Sarkar et al., 2005; Serrano et al., 2009). Natural data also often has the property that the marginal distribution across entities is highly skewed, following a Zipfian (power law) distribution with a long tail of infrequent items (Piantadosi, 2014; Smith et al., 2018; Zipf, 1949). Finally, the 'meaning' of entities in natural data (such as words in natural language) is often dynamic rather than fixed. That is, a single entity can have multiple possible interpretations (polysemy and homonymy, in language) and multiple entities can map to the same interpretation (synonymy, in language), usually in a context-dependent way. The combination

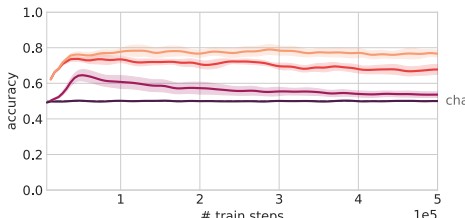
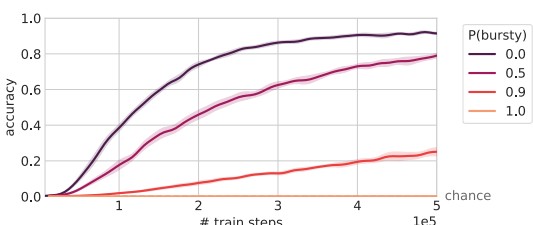

Figure 2: Effects of burstiness. $P(\text{bursty})$ indicates the proportion of training sequences that were bursty vs non-bursty, and models are evaluated on the two types of evaluation sequences, over the course of training. Burstiness in the training data increases in-context learning, and decreases in-weights learning. Also, over the course of training, in-context learning tends to decrease while in-weights learning increases.

of these properties may result in training data that occupies some middle-ground between the data used in canonical supervised learning and that used for few-shot meta-training.

In particular, standard supervised training typically consists of item classes that recur with uniform regularity, and with item-label mappings that are fixed throughout training – these properties allow a model to gradually learn over time, by encoding information into its weights, e.g. via gradient descent. By contrast, few-shot or in-context meta-training generally involves training a model directly on specially crafted sequences of data where item classes only recur and/or item-label mappings are only fixed *within episodes* – they do not recur and are not fixed across episodes (Santoro et al., 2016; Vinyals et al., 2016). Naturalistic data, such as language or first-person experience, has characteristics of both of these data types. As in supervised training, items (words) do recur, and the relationship between an entity and its interpretation (or meaning) is fixed, to some degree at least. At the same time, the skewed and long-tailed distribution of natural data means that some entities recur very frequently while a large number recur much more rarely. Importantly, however, these rare items are often bursty, making them disproportionately likely to occur multiple times within a given context window, somewhat like a sequence of 'meta-training' data. We can also see the dynamic relationship between entities and their interpretation (epitomized by synonyms, homonyms, and polysemy, in the case of language) as weaker versions of the completely dynamic item-label mappings that are used in few-shot meta-training, where the mappings are randomly permuted on every episode.

In this paper, we experimentally manipulated the distributional properties of the training data and measured the effects on in-context few-shot learning. We performed our experiments over data sequences sampled from a standard image-based few-shot dataset (the Omniglot dataset; Lake et al., 2019). At training, we fed each model (such as a transformer or recurrent network) with input sequences of Omniglot images and labels, varying the natural data-inspired distributional properties of choice. At evaluation, we assessed whether these properties gave rise to in-context learning abilities.

Our results showed that, indeed, in-context learning emerges in a transformer model only when trained on data that includes both burstiness and a large enough set of rarely occurring classes.We also tested two instantiations of the kinds of dynamic item interpretation observed in natural data – having many labels per item as well as within-class variation. We found that both interventions on the training data could bias the model more strongly towards in-context learning. The models we tested typically exhibited a tradeoff between rapid in-context learning vs. relying on information that was stored through slow, gradient-based updates ('in-weights" learning). However, we found that models could simultaneously exhibit *both* in-context learning and in-weights learning when trained on a skewed marginal distribution over classes (akin to the Zipfian distribution of natural data).

At the same time, architecture is also important. Unlike transformers, recurrent models like LSTMs and RNNs (matched on number of parameters) were unable to exhibit in-context learning when trained on the same data distribution. It is important to note, however, that transformer models trained on the wrong data distributions still did fail to exhibit in-context learning. Thus, attention is not all you need – architecture and data are both key to the emergence of in-context learning.

(a) In-context learning on holdout classes.    (b) In-weights learning on trained classes.

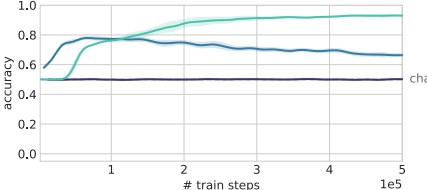 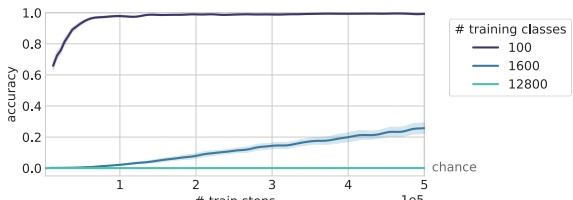

Figure 3: Effects of number of classes. Increasing the number of training classes improves in-context learning, while reducing in-weights learning.

## 2 Experimental Design

### 2.1 The training data

To investigate the factors that lead to in-context few-shot learning, we created training and evaluation sequences using the Omniglot dataset (Lake et al., 2019, MIT License), a standard image-label dataset for few-shot learning. Omniglot consists of 1623 different character classes from various international alphabets, with each class containing 20 handwritten examples. Using the Omniglot dataset allowed us to apply evaluation procedures that are standard in the study of few-shot learning. The few-shot challenge is to classify an example of a character class that was never seen in training, based only on a few examples of that class and some alternate classes.

The training data consisted of sequences of images and labels (Fig 1b). The first 16 elements of each sequence comprised the 'context', and consisted of 8 image-label pairs (where each image was always followed immediately by its corresponding label). The final element was the 'query' image, and the aim of the model was to predict the correct label for the query.

Images were allowed to recur throughout training, and the integer label for each image class was unique and fixed across training, as in typical supervised datasets. We emphasize that this is a major departure from conventional few-shot training, where item-label mappings are completely novel on each episode, or the items themselves are novel on each episode.

In our standard experiments, we trained the model on a mixture of 'bursty' and 'non-bursty' sequences. In the bursty sequences, the query class appeared 3 times in the context. There are many possible ways to quantify or instantiate burstiness (e.g., Altmann et al., 2009; Alvarez-Lacalle et al., 2006; Lambiotte et al., 2013; Neuts, 2007; Sarkar et al., 2005; Serrano et al., 2009), but the 'bursty' sequences in our experiment were designed to reflect the within-context burstiness that is observed in e.g. language. To prevent the model from simply outputting the most common label in the sequence, a second image-label pair also appeared 3 times in the context. For the non-bursty sequences, the image-label pairs were drawn randomly and uniformly from the full Omniglot set. We can continuously vary the overall degree of burstiness in a dataset by changing the proportion of 'bursty' vs 'non-bursty' sequences.

### 2.2 The model

Each element of a sequence was first passed through an embedder (a standard embedding layer for the integer labels, and a ResNet for the images; He et al., 2015). These embedded tokens were passed into a causal transformer model (Fig 1a) (Vaswani et al., 2017). Unless stated otherwise, we used a transformer with 12 layers and embedding size 64. The model was trained on a softmax cross-entropy loss on the prediction for the final (query) image.

### 2.3 The evaluation data

We evaluated trained models on two types of sequences, to measure (1) in-context learning and (2) in-weights learning. As in the training sequences, the evaluation sequences also consisted of 8 pairs of 'context' image and label tokens, followed by a single 'query' image token.

To measure a trained model's ability for in-context few-shot learning, we used a standard few-shot setup. The context consisted of a random ordering of 2 different image classes with 4 examples each, and the query was randomly selected from one of the two image classes (a '4-shot 2-way' problem, in few-shot nomenclature). Unlike in training, where the labels were fixed across all sequences, the labels for these two image classes were randomly re-assigned for each sequence. One image class was assigned to 0, and the other to 1 (Fig 1c). Because the labels were randomly re-assigned for each sequence, the model must use the context in the current sequence in order to make a label prediction for the query image. Unless stated otherwise, in-context learning was always evaluated on holdout image classes that were never seen in training.

Although the model is always required to perform a full multi-class classification over all possible output labels (as in training), few-shot accuracy is computed by considering the model outputs only for the two labels seen in the few-shot sequence (0 and 1), with chance at $1/2$. This ensures that performance above chance cannot be due to e.g. randomly selecting one of the labels from the context. Note also that the model was evaluated for in-context learning on novel image classes, but not novel labels (see the appendix for further discussion).

To measure in-weights learning of trained classes in a model, evaluation sequences consisted of image classes that were selected uniformly without replacement, with the same labels that were used in training (Fig 1d). Because the image classes were forced to be unique within each sequence, the query had no support in the context. Thus, the only way for a model to correctly predict the label was to rely on information stored in the model weights. For this problem, where the correct query label could be any of the labels seen in training, chance was usually $1/1600$.

## 3 Results

### 3.1 What kinds of training data promote in-context learning?

**Burstiness.** In our first experiments, we vary levels of burstiness in the training data by varying the proportion of bursty vs non-bursty sequences in the training data (as described in Section 2.1). These experiments replicate the finding that transformers can acquire in-context few-shot learning even without explicit meta-training. They further show that, as hypothesized, the model displays better in-context learning with more burstiness in training (Fig 2a). We also see that in-context learning trades off against in-weights learning – greater burstiness simultaneously leads to lower weight-based learning (Fig 2b). Interestingly, the models can in some cases lose an initial bias towards in-context learning, moving towards in-weights learning over the course of training.

**A large number of rarely occurring classes.** Our second set of experiments show that in-context learning performance depends on the number of training classes (keeping the level of burstiness fixed at $p(\text{bursty}) = 0.9$). As we increase the number of classes from 100 to 1600 (and correspondingly decrease the frequency of each class), we see improvement of in-context learning (Fig 3a). As before, we also see an accompanying decrease in in-weights learning (Fig 3b). This accords with our hypothesis about the importance of having a long tail in the distribution, or a large vocabulary. Note that the bias against in-weights learning cannot be explained by the number of exposures to each class – even controlling for the number of exposures, the model trained with 1600 classes is much slower to achieve similar levels of in-weights learning. Importantly, we need both burstiness and a large number of classes for in-context learning to emerge. In order to further increase the number of classes beyond the 1623 available in the original Omniglot dataset, we rotated ($0°, 90°, 180°, 270°$) and flipped (left-right) the images, obtaining $8\times$ more image classes. We ensured that the holdout set did not include transformed versions of train images. Training on these 12800 classes further improved in-context learning (and reduced in-weights learning) (Fig 3). However, some images in Omniglot have rotational or mirror symmetries, so that the models trained on 12800 classes may additionally be pushed towards in-context learning by a label-multiplicity effect, described next.

**Multiplicity of labels.** Our third set of experiments explored the effect of dynamic meanings, with training distributions where images did not have completely fixed labels. Each image class was assigned to multiple possible labels and, in the data sequences, the label shown after each image was randomly selected among the possible labels. If a class appeared more than once in the same sequence, the label was consistent for all presentations within that sequence (this is commonly the case in natural data such as language, too; Gale et al., 1992). In Fig 4, we see that increasing the 'label

multiplicity' (the number of labels per class) also increases in-context learning. Again, burstiness was fixed for these experiments at $p(\text{bursty}) = 0.9$.

(a) In-context learning on holdout classes.

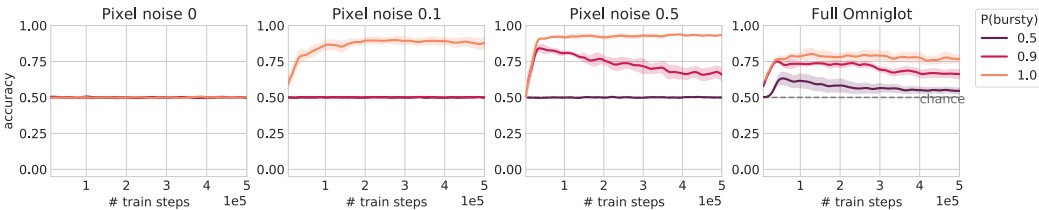

(b) In-weights learning on trained classes.

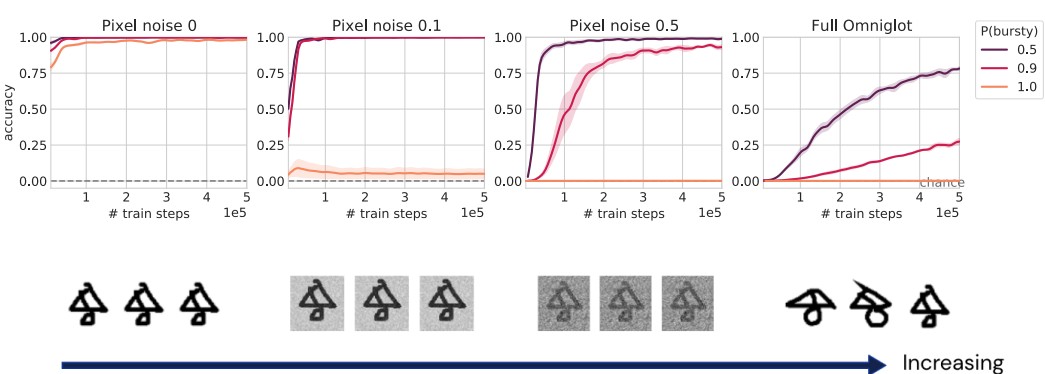

Figure 5: Effects of within-class variation. When we increase the within-class variation (from left to right), in-context learning tends to increase (a) while in-weights learning decreases (b). Both effects are nonetheless upper-bounded by the difficulty of within-class generalization, with the 'Full Omniglot' problem being more difficult than the rest. For the 'Full Omniglot' experiments, each class contained the full set of 20 Omniglot exemplars per class. For the remaining experiments, each consisted of only a single Omniglot exemplar image, with varying levels of Gaussian pixel noise.

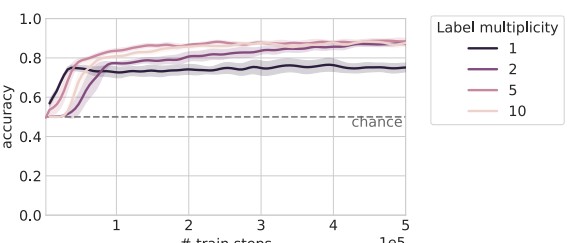

Figure 4: Dynamic meanings improve in-context learning. Increasing the number of labels per class ('label multiplicity') increases in-context learning.

**Within-class variation.** We then explored another source of dynamic variation of meaning – the amount of variation within image classes themselves. In the lowest-variation condition, each image class consists of only a single image, i.e. the images for a given class were always identical. In the medium-variation conditions, we added Gaussian pixel noise to the images (resampled for each presentation). In the high-variation condition, we used the full Omniglot classes (each class consists of 20 different images drawn by 20 different people). To our surprise, we found that greater within-class variation leads to greater in-context learning (Fig 5). In other words, making the generalization problem harder actually made in-context learning emerge more strongly – it preferentially hampered in-weights learning more than it hampered in-context learning.

Across all the above experiments (Figs 2-5), we also evaluated in-context learning on training classes (rather than holdout classes), again randomly assigning the classes to labels 0 and 1 (rather than using the ones seen in training). Evaluations looked similar in all cases, with only slightly higher performance (Appendix C.3).

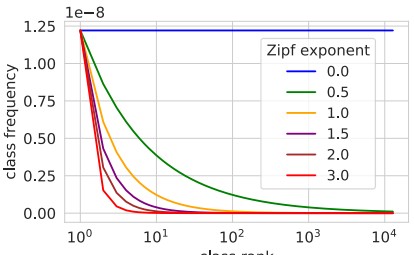
(a) Examples of Zipfian distributions.

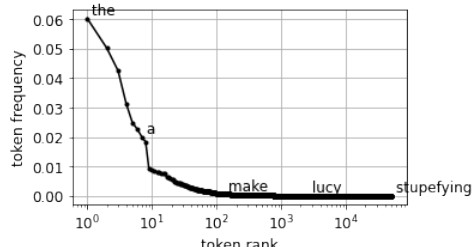
(b) Distribution of tokens in a natural language corpus.

(c) In-context learning on holdout classes.

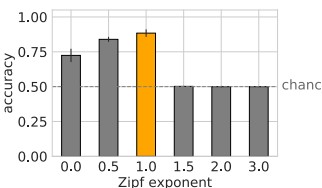

(d) In-weights learning on common classes.

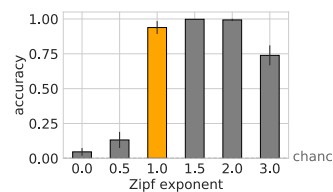

(e) In-weights learning on rare classes.

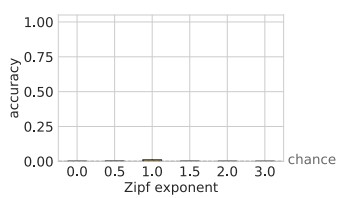

Figure 6: Effects of training on Zipfian (rather than uniform) marginal distributions over classes. (**a**) Examples of Zipfian distributions with varying exponents. (**b**) The distribution of tokens in an example English-language corpus. In (**c**-**e**), bars indicate mean evaluation accuracy in the window [400k, 500k] steps of training. (**c**) As we increase the Zipf exponent, i.e. increasing the skew on the class distribution, we see a decrease in in-context learning. (**d**) In-weights learning of the 10 most common classes, in contrast, increases with more skew. With uniform training (Zipf exponent = 0), the model exhibits only in-context learning and not in-weights learning. However, if we train on skewed distributions, there is a sweet spot where both in-context learning and in-weights learning can be maintained at a high level in the same model (Zipf exponent = 1, for this particular training regime). Coincidentally, a Zipf exponent of 1 corresponds approximately to the skew in many natural languages. (**e**) Rare items from training are never memorized (performance is at chance for all Zipf exponents).

## 3.2 What kinds of training data enable in-context learning and in-weights learning to *co-exist* in the same model?

In the previous section, we saw a consistent tradeoff between in-context learning and in-weights learning – no models could maintain both. However, it is useful for a model to have both capabilities – to remember information about classes that will re-appear in evaluation, while also being able to perform rapid in-context learning on new classes that appear only in holdout. Large language models certainly do have both of these capabilities. How might we achieve this?

For all prior experiments, the training data were marginally distributed uniformly over classes, even if the data were non-uniform in other ways. I.e., each class was equally likely to appear, marginalizing across the dataset. We postulated that we might achieve both types of learning in the same model by instead training on marginally-*skewed* distributions. In this case, some classes appear very commonly, while most classes appear very rarely. Many natural phenomena such as word distributions take this form, and are classically described as a Zipfian (power law) distribution (Zipf, 1949):

$$p(X = x) \propto \frac{1}{x^\alpha} \tag{1}$$

Here, $X$ is the rank of the class (e.g. 1 for the most common class), and the exponent $\alpha \in [0, \infty)$ determines the degree of skew. Fig 6a shows some examples of Zipfian distributions with various exponents. Fig 6b shows an example of token distributions in English (from the Brown corpus; Francis and Kucera, 1979).[2] This type of skewed distribution could allow a model to learn common classes in its weights, while the long tail of rare classes simultaneously induces an ability for in-context learning.

---

[2]Plot generation adapted from https://gist.github.com/fnielsen/7102991

To test this hypothesis, we trained on Zipfian distributions, varying the Zipf exponent and hence the degree of skew. We used the same training sequences as before, with 12800 classes and $p(\text{bursty}) = 0.9$. Our results are shown in Figs 6c-e. We evaluate in-weights learning separately on common classes (the 10 classes seen most often in training) and on rare classes (the remaining classes). When there is no skew, all classes are relatively rare, and we see high levels of in-context learning but no in-weights learning. Increasing the skew leads to the loss of in-context learning and increased in-weights learning of common classes.[3] In between the two extremes, we observe a sweet spot at Zipf exponent = 1, where the model maintains high levels of both in-context learning and in-weights learning of common classes. Intriguingly, natural languages are best described by a Zipfian distribution with an exponent of approximately 1 (Piantadosi, 2014). Note though that the sweet spot for simultaneously maintaining in-weights and in-context learning in transformers may differ, depending on the training regime.

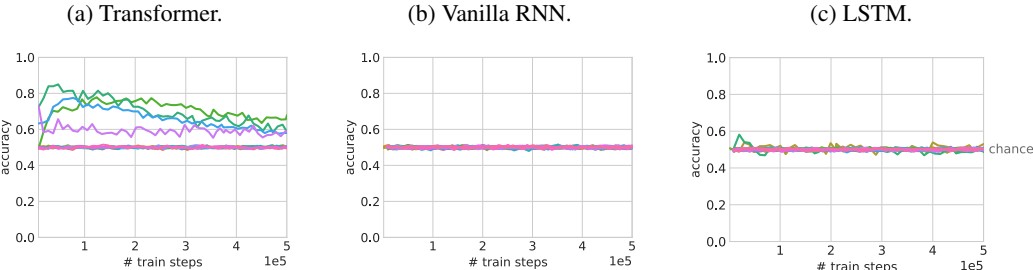

| (a) Transformer. | (b) Vanilla RNN. | (c) LSTM. |

Figure 7: In-context learning in transformers vs. recurrent architectures. We compare architectures while holding fixed the number of layers, hidden layer size, and number of parameters. Only a transformer is able to attain in-context learning; the Vanilla RNN and LSTM never perform above chance. One run was performed for each set of hyperparameters in a hyperparameter sweep. Each color denotes one run, but not any particular hyperparameter values.

### 3.3 But architecture does matter too.

To investigate whether these results are specific to transformer models, we performed similar experiments using recurrent sequence models. For these models, we simply replaced the transformer with either a vanilla recurrent neural network (RNN; David E. Rumelhart et al., 1985) or a long short-term memory network (LSTM; Hochreiter and Schmidhuber, 1997). We used the same training sequences as before, with 1600 classes and $p(\text{bursty}) = 0.9$. We also used the same image and label encoders, and cross-entropy classification loss. The recurrent models were matched to the transformer for depth, number of parameters, and hidden layer size. We performed a comprehensive hyperparameter search for all models (see Appendix for details).

In these experiments, we see that the recurrent models are never able to achieve in-context learning, despite the parity in training setup (Fig 7). Interestingly, the transformer actually outperforms the recurrent models on in-weights learning as well (see Fig 8 in the Appendix), indicating that we cannot explain these results by proposing that recurrent models are simply more biased towards in-weights learning than transformers.

## 4 Discussion

In summary, we find that both data and architectures contribute significantly to the emergence of in-context learning in transformers.

**Data properties that promote in-context learning.** We identify several features of training data that can promote in-context learning – burstiness, number and rarity of training classes, and dynamic meaning (as instantiated by multiple labels per class or within-class variation). These data properties allow in-context learning to emerge despite differing significantly from the data used in standard few-shot meta-training, in that we allow items and item-label mappings to recur throughout training. These properties are also central features of natural data including language, and thus may explain the remarkable emergence of in-context learning in large language models without explicit meta-training.

---

[3]We see decreased in-weights learning for Zipf exponent = 3, because that level of skew leads to extreme focus on a tiny number of classes (e.g. the three most common classes form 97% of the data).

**Effects of architecture.** We find that architecture does matter as well. Transformers show a significantly greater capacity for in-context learning than recurrent models – we were completely unable to elicit in-context learning in recurrent models, even with the training procedure, number of parameters, and model architecture otherwise matched to the transformer experiments. We emphasize however that the transformer architecture alone was insufficient for eliciting in-context learning – it was necessary for the training data to exhibit at least burstiness and large numbers of classes, too.

**In-context vs. in-weights learning.** In most cases, we found that transformers exhibited a tradeoff in their bias towards either in-context learning or in-weights learning, and could not maintain both in the same model. We characterize this behavior as a 'bias', because neither type of learning is 'correct' per se. For our training data, an in-context learning strategy and an in-weights learning strategy will give the same answer, since the labels are fixed. Thus, in the in-context evaluation sequences, it is ambiguous (by design) whether the model should use the labels seen in training or in the current context, allowing us to measure the model's bias. We also note that even models with an initial bias towards in-context learning can often move towards in-weights learning with enough repetition.

However, it is often important and useful for a model to exhibit both capabilities – to perform slow, gradient-based in-weights learning of class information that is presented during training, while also being able to quickly learn (without weight updates) about new classes that appear only in evaluation. Indeed, large language models exhibit both of these capabilities (Brown et al., 2020). In our experiments, we discovered that an additional language-like distributional property could allow models to maintain both capabilities as well – a skewed, Zipfian distribution over classes. This allowed the models to retain information in their weights about common classes, while simultaneously developing in-context learning abilities that were presumably induced by the long tail of rare classes.

**Implications for understanding language models.** Our findings have a few noteworthy implications. First, by pointing to specific distributional properties of training data that both exist in language and also promote in-context learning, these results may help us reach a more scientific understanding of why in-context learning emerges in transformer-based language models. This is an area of increasing interest (e.g. Min et al., 2022; Razeghi et al., 2022; Webson and Pavlick, 2021; Xie et al., 2021).

We emphasize that the transformers in our experiments successfully performed in-context evaluation on *holdout* classes, and only performed slightly better with in-context evaluation on trained classes. These results are counter to an emerging narrative that large language models may not actually be performing genuine in-context learning, and simply draw on examples seen in training (Min et al., 2022; Razeghi et al., 2022; Xie et al., 2021) – our experiments show that naturalistic distributional properties can give rise to a capacity for in-context learning on classes that were never seen in training.

**Broader implications.** This understanding may also help us design and collect datasets to achieve in-context learning in domains *outside* of language, an area of ongoing research (e.g. Finn et al., 2017; Hill et al., 2020; Wang et al., 2016). Given that reinforcement learning environments are generally designed to be uniformly distributed (Chan et al., 2022), or that supervised datasets are frequently rebalanced to have *more uniform* distributions (Chawla et al., 2002; Katharopoulos and Fleuret, 2019; Van Hulse et al., 2007), we may be missing an opportunity to endow non-language models with a powerful capability. We may need to consider data distributions more carefully when pre-training in non-language domains, as well. For example, recent work has shown that pre-training on language data was useful for offline reinforcement learning, but pre-training on vision data was not (Reid et al., 2022) – could this difference be due to the non-uniform, structured distribution of the language data?

**Cognition and neuroscience.** Our experiments could also potentially inspire research on the role of non-uniformity in human cognitive development. Infants rapidly learn statistical properties of language (Saffran and Kirkham, 2018) — could these distributional features help infants to acquire an ability for rapid learning, or serve as useful pretraining for later learning? And could non-uniform distributions in other domains (e.g., vision) also contribute to this development (cf. Smith et al., 2018)?

Our results may also relate to complementary learning systems theory (Kumaran et al., 2016; McClelland and O'Reilly, 1995) and its application to language understanding in the brain (McClelland et al., 2020). According to this theory, the neocortical part of the language system bears similarities to the weights of neural networks, in that both systems learn gradually through the accumulated influence of large amounts of experience. The hippocampal system plays a role similar to the context window in a transformer model, by representing the associations encountered most recently (the hip-

pocampus generally has a time-limited window; Squire, 1992). [4] In this light, it is possible to see the human hippocampal system as a system that provides the architectural advantage of the transformer's context representations for in-context learning.

**Future directions.** The above results suggest exciting lines of future research. How do these data distributional properties interact with reinforcement learning vs. supervised losses? How might results differ in experiments that replicate other aspects of language and language modeling, e.g. using symbolic inputs, training on next-token or masked-token prediction, and having the meaning of words determined by their context? For models that display both in-context and in-weights learning, it would be interesting to understand contextual cuing of already learned information – does this increase with more exposure? There is also a lot more to understand about the behaviors and biases of transformers vs. recurrent architectures – why do transformers seem to be more capable of in-context learning?

**Non-uniformity.** Finally, we hope to emphasize the dual nature of non-uniformity in training data. While it can impair both supervised and reinforcement learning (Chan et al., 2022; Van Hulse et al., 2007), we show here that non-uniform training distributions can induce the emergence of at least one useful and interesting capability, and thus can be an opportunity as well as a challenge.

## Acknowledgments and Disclosure of Funding

We would like to thank the following colleagues for invaluable feedback and discussion: Kris Cao, Toni Creswell, Kevin Miller, Ivana Kajic, Andrea Banino, Ishita Dasgupta, Kenneth Marino, Irina Higgins, Murray Shanahan, Kyriacos Nikiforou, Richard Evans, Christos Kaplanis, David Reichert, Dave Abel, and Drew Hudson.

We would also like to thank the following colleagues for their contributions to the transformer implementation: Igor Babuschkin, Junyoung Chung, David Choi, Tamara Norman, Sebastian Borgeaud, Jack Rae, David Saxton, Yujia Li, Phil Blunsom, Maribeth Rauh, Roman Ring, Nate Kushman, Vinicius Zambaldi, Tom Hennigan

This work was funded by DeepMind.

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
