# Appendix

## A   Model and training procedure: details

All experiments used the same model and training procedure, unless stated otherwise. The transformer consisted of 12 layers, with embedding dimension 64 and 8 heads. The images were embedded by a ResNet with two blocks per group and channels per group (16, 32, 32, 64), and which was not pre-trained. The integer labels were embedded using a standard embedding layer. The input embeddings were augmented with a standard sinusoidal positional encoding. Experiments were run for 500k training steps on 16 TPU v2 or v3 cores. They were trained using Adam and a learning rate schedule with a linear warmup up to a maximum learning rate of 3e-4 at 4000 steps, followed by an inverse square root decay. The experiments shown in Figs 5 and 6 were run with 3 seeds each (because of the larger number of conditions in those experiments), and all other experiments were run with 5 runs each. In all figures, (shaded) error bars indicate standard deviation around the mean.

## B   Possible extensions: Generating new image labels

An important constraint of the model implementation and evaluation procedure is that we do not require the models to handle novel image labels, only novel image classes. Thus, in-context learning is evaluated on labels that were previously seen in training, i.e. 0 and 1 (on the Zipfian-skewed experiments, these corresponded to two most common labels). Note that, if anything, this causes in-context learning to be more difficult for the model, since it must overcome existing image-label associations that were learned in training.

However, as future extensions, it would be possible to extend the model to handle novel labels as well. For example, we might tie the input and output embedding layers (sometimes done in large language models, though mainly for computational efficiency), or to generate novel labels as combinations of already-seen tokens (akin to language models that use the SentencePiece family of tokenization).

## C   Experiments comparing recurrent vs. transformer

### C.1   Architectural details

Hyperparameter sweep:

- Max learning rate: 15 samples log-uniform distributed over the range [1e-5, 0.1]
- Num warmup steps: 15 samples log-uniform distribution over the range [1, 10000]

We performed 15 runs for each architecture (Transformer with 2 or 12 layers, LSTM with 2 or 12 layers, Vanilla RNN with 2 or 12 layers), i.e. 90 runs total.

Parameter counts:

- Transformer with 12 layers: 831,479
- LSTM with 12 layers: 627,959
- Transformer with 2 layers: 331,639
- LSTM with 2 layers: 297,719

### C.2   In-weights learning

Transformers exhibited similar or slightly higher in-weights learning than the recurrent models (Fig 8), indicating that their superior in-context learning performance (as seen in Fig 7) cannot simply be explained by a bias towards in-context learning and against in-weights learning.

### C.3   In-context evaluation on trained classes

Fig 9 shows results of evaluating in-context learning on classes that were seen in training, rather than on holdout classes (the standard evaluation setting for few-shot learning, as described in Sec 2.3. The

pattern of results is very similar between the two settings, with just slightly higher performance when evaluating on training classes.

## C.4 Multi-class in-context evaluation

For completeness, we also report the in-context evaluation results by computing accuracy fully multi-class across all possible outputs of the model (Fig. 10). This is in contrast to the evaluations that were reported in the main text (as described Sec 2.3), across just the two labels that appeared in context; the two-choice evaluation provides a more sensitive measure of performance, ensuring that all experimental conditions have the same levels of chance, and also ensuring that the model cannot achieve above-chance performance simply by randomly selecting from the labels in context. Note that, across training and both types of evaluation (in-context and in-weights), the model is the same – it is trained to perform multi-class classification.

Multi-class evaluation shows the same patterns of results as the two-way evaluation from the main text. Note that the multi-class evaluation results showing the effects of the number of training classes (Fig 10b) and dynamic meanings (Fig 10d) need to be interpreted with caution, because the number of model outputs changes in the different conditions, so that task difficulty and chance levels differ for each.

The multi-class evaluation uncovers one additional interesting result, for the models trained on Zipfian distributions (Fig 10e). As in the two-choice evaluation setting, a Zipfian distribution with an exponent of 1 is the only one able to elicit significantly above chance accuracy on both in-context evaluation and in-weights evaluation on common classes. However, Zipf 1 models have relatively lower few-shot performance when evaluated in the fully multi-class setting. Further investigation revealed that this was because those models have overall less tendency to output labels from context (Fig 11). Nonetheless, the Zipf 1 models do perform significantly above chance in both settings, and when forced to choose between the two labels that are shown in context, the model performs very well on these sequences. This indicates that, interestingly, a model can attain both in-weights and in-context learning abilities and process an input sequence in both ways, even if it is unsure which of those two processes it should output the result for.

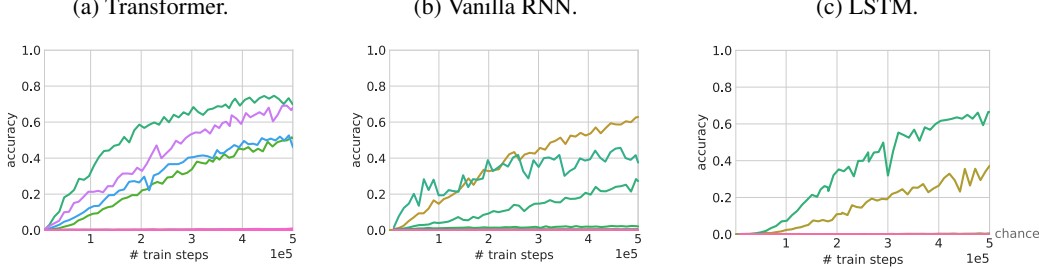

Figure 8: In-weights learning in transformers vs. recurrent architectures. We compare architectures while holding fixed the number of layers, hidden layer size, and number of parameters. One run was performed for each set of hyperparameters in a hyperparameter sweep. Each color denotes one run, but not any particular hyperparameter values.

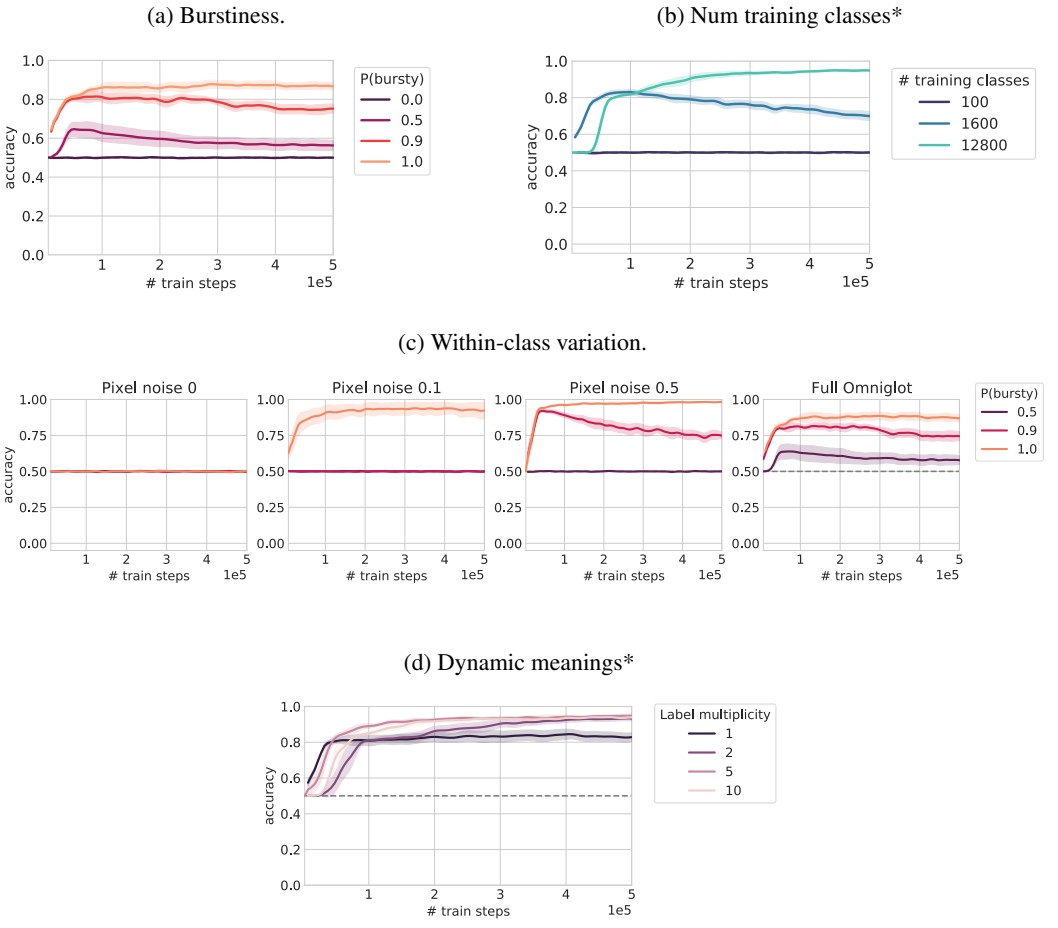

Figure 9: In-context learning accuracy, evaluated on classes that were observed in training, rather than holdout classes. Patterns of results are very similar to those shown in the main text, with overall slightly higher performance when evaluated on training classes.

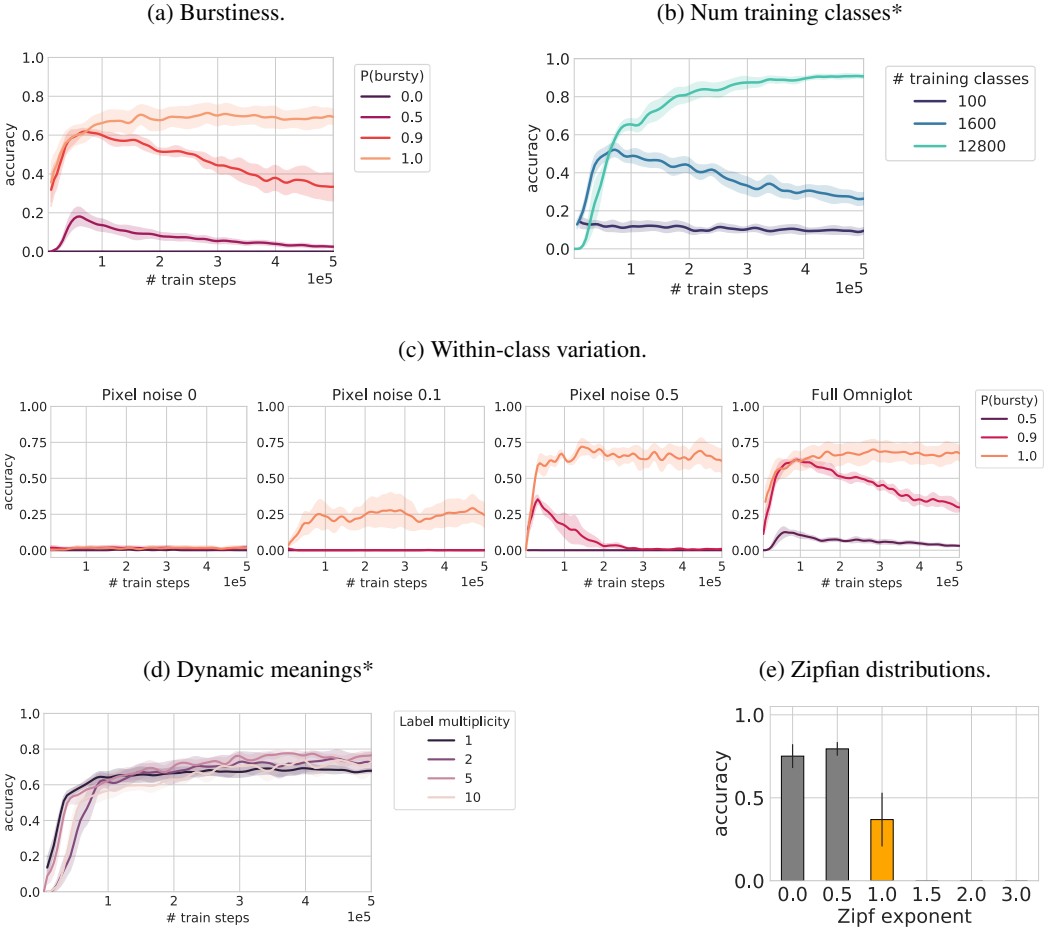

Figure 10: In-context learning accuracy, evaluated fully multi-class across all possible outputs of the model, rather than considering outputs on just the two labels that appeared in context. Patterns of results are qualitatively similar to those shown in the main text. *Figures (b) and (d) should be interpreted with caution, because the total number of classes differ for each experimental condition, and therefore chance levels. We include them for completeness.

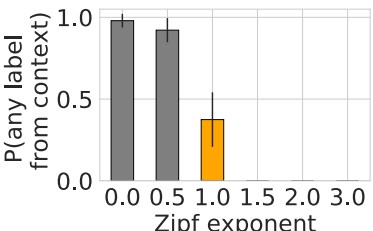

Figure 11: Frequency of outputting any of the two labels that appear in context, for a model trained on Zipfian distributions and evaluated on in-context evaluation sequences.