# OpenReview forum: "Data Distributional Properties Drive Emergent In-Context Learning in Transformers"
_NeurIPS.cc/2022/Conference — NeurIPS 2022 Accept_

### Official Review · Reviewer_EeoY · 2022-07-04

**Rating:** 8
**Confidence:** 3
**Soundness:** 2 fair
**Presentation:** 3 good
**Contribution:** 4 excellent

**Summary:**

Recently, the emergence of in-context learning raises a question about the cause of this learning behavior. This paper considers in-context few-shot learning for image data. In the setting, this paper finds that in-context learning emerges when the training data is bursty. In addition, when the data follows a skewed Zipfian distribution, both in-context learning and in-weight learning can co-exist in a single model.

Updates after seeing other reviews: I under-estimated the core contributions of this paper, both in the novelty of the perspective and the setting of empirical evaluations. However, I still think this paper will benefit from more rigorous descriptions of some key terms. I increased the score to 5 (confidence rating 3).

**Questions:**

- It appears that burstiness is treated as a binary feature (defined in Figure 1 caption, lines 5-7). This is not a rigorous definition, and would definitely benefit from additional elaborations.
- Similarly, "in-weight learning" and "in-context learning" are also defined in Figure 1 caption (lines 6-7), which are not quantitative definitions. The in-weight vs. in-context learning is an effect mentioned a lot in this paper. I think a less ambiguous definition than just "learning labels across sequences" or "referring back to the context" would be necessary.
- "Bursty sequences can be solved by in-weights learning or in-context learning". So the non-bursty sequences can not be solved by either in-weights learning or in-context learning?
- How does in-context learning work together with few-shot learning? Did you sample multiple demonstration sequences and run few-shot learning? Compared to few-shot learning (using just one image as input), this setting has a lot of computational overheads. Compared to in-context learning (without changing model parameters), this approach is limited to much smaller models. What are the benefits of in-context few-shot learning?
- Sec 3.1: Doesn't bustiness overlap to some extent with "a large number of rarely occurring classes"?

**Limitations:**

The limitations are adequately addressed.

**Strengths And Weaknesses:**

**Strengths**

- The study of interaction between in-context learning and in-weight learning is novel.
- The inquiry into the reason that lead to in-context learning is important.
- The experiments show some interesting findings.

**Weaknesses**

- Formatting: The page limit is 9 pages right? Lines 314-317 appear in page 10.
- Three key concepts discussed in this paper, in-context learning, in-weight learning, and burstiness, lack quantified definitions. Please refer to my comments below.

---

> ### Author Response · Authors · 2022-08-02
> **Thank you for your thoughtful review (part 1)**
>
> Dear Reviewer,
>
> Thank you for your thoughtful review of our paper, and for your evaluation that our paper is novel, interesting, and important.
>
> Thank you also for your pointers for clarification. We appreciate the opportunity to further improve our exposition, and we have addressed your comments below and with updates to the paper.
>
> –
>
> [[ RE: A definition for burstiness ]]
>
> The fields of statistics, computer, and linguistics have proposed many ways to quantify burstiness, e.g. as described in the references included in the introduction, but there is unfortunately no single standard definition. A small sampling of of these include: a "numerical discrepancy" (the difference in counts between the current context and the mean), the Fano factor (a ratio between the variance and mean of counts), or the degree of skew when fitting an exponential distribution to inter-event intervals. Rather than choosing a restrictive definition, we wanted to emphasize the general phenomenon, which is observed across many natural domains. We therefore described burstiness in the introduction as simply "a distribution that is not uniform across time, instead tending to appear in clusters", in order to capture this general phenomenon. Wikipedia states, similarly, that "In statistics, burstiness is the intermittent increases and decreases in activity or frequency of an event." For our experiments, we use the functional definition that bursty sequences have multiple occurrences of the query category in the context, chosen to reflect one type of burstiness that can be observed in e.g. language (within a single context window). We have added emphasis on these points in the section "The training data". Even in our particular instantiation, however, burstiness is a continuous feature of the overall dataset, which we vary by increasing or decreasing the proportion of "bursty" sequences. We have clarified these points in the paper. We have also added emphasis on the wide variety of possible ways to quantify burstiness, and additional references.
>
> [[ RE: Definitions for "in-weights" vs "in-context" learning ]]
>
> The definitions for these terms can be found in the main text, rather than in the figure captions. We previously defined "in-weights learning" as "information that is stored through slow, gradient-based updates". We defined "in-context learning" as "few-shot learning from a few examples without the need for any weight updates … referred to as 'in-context learning', as the output is conditioned on the context". We believe that these are in line with standard definitions for these terms. However, we have now rewritten the first paragraph of the introduction to make these definitions much more explicit:
>
> "Large transformer-based language models show an intriguing ability to perform in-context learning \citep{brown_language_2020}. This is the ability to generalize rapidly from a few examples of a new concept on which they have not been previously trained, without gradient updates to the model. In-context learning is a special case of few-shot learning in which the output is conditioned on examples from a 'context', and where there are no gradient updates. It contrasts with 'in-weights' learning, which is the standard setting for supervised learning – this is slow (requiring many examples), and depends on gradient updates to the weights."
>
> We have also emphasized the defined terms in bold, to draw readers' attention and make them easier to refer back to. Please do let us know if further clarification is needed.
>
> [[ RE: "Bursty sequences can be solved by in-weights learning or in-context learning". So the non-bursty sequences can not be solved by either in-weights learning or in-context learning? ]]
>
> Non-bursty sequences can only be solved by in-weights learning.
>
> (continued below in part 2)

---

> > ### Author Response · Authors · 2022-08-02
> > **Thank you for your thoughtful review (part 2)**
> >
> > [[ RE: How does in-context learning work together with few-shot learning? Did you sample multiple demonstration sequences and run few-shot learning? Compared to few-shot learning (using just one image as input), this setting has a lot of computational overheads. Compared to in-context learning (without changing model parameters), this approach is limited to much smaller models. What are the benefits of in-context few-shot learning? ]]
> >
> > We are not completely certain that we understand this question, but we think that there may potentially be a misunderstanding of the definitions of few-shot learning and in-context learning. Hopefully the more explicit definitions added to the introduction will ameliorate this. Few-shot learning generally involves learning from a *few* image-label pairs, rather than one. In-context learning is a specific type of few-shot learning, where there are no gradient updates, and where the model output is conditioned on a "context" that is provided at evaluation.
> >
> > In the second part of your question, we believe you are pointing out that few-shot learning *with* gradient updates is an approach that is limited to smaller models, perhaps because gradient updates are expensive for a larger model? This is true, and it is very interesting to consider your question about the benefits of in-context learning vs few-shot learning with gradient updates. In-context learning is beneficial because it is e.g. far less computationally expensive, and able to deal with non-uniform sampling across sequences. Note that our paper is not intended to advocate for in-context learning, but rather to explain the factors that drive this interesting emergent phenomenon.
> >
> > [[ RE: Sec 3.1: Doesn't bustiness overlap to some extent with "a large number of rarely occurring classes"? ]]
> >
> > We vary these distributional properties independently in our experiments. The number of rarely occurring classes is a property describing the number of possible images that can appear; but given a fixed distribution across all images, it is possible to make that distribution bursty (by packing images into the same sequence) or distributed. As a toy example, if the distribution were “a” x 2 and “b” x 2, this could be made into bursty sequences “aa” and “bb”, or into non-bursty sequences “ba” and “ab”. The frequency of "bursty" sequences can be controlled independently of the number of classes.
> >
> > We hope that these clarifications address your questions. Thank you again for your review, and please do let us know if you have any additional questions.
> >
> > Best regards,
> >
> > The authors

---

> > > ### Comment · Reviewer_EeoY · 2022-08-02
> > > **Thank you for the clarifications**
> > >
> > > Thank you for clarifying the points. Now I understand more about what this paper is trying to do. This paper indeed opens up a really important research direction -- where do the in-context learning ability come from -- and provides some analysis towards an answer, in-context learning ability comes from the data distributional properties. To fully support this answer (e.g., which property in the data distribution allows the in-context learning, and if any of the property interacts with the structural choices), tens of subsequent papers are needed. I'll update my score.
> > > Best regards,
> > > Reviewer EeoY

---

> > > > ### Author Response · Authors · 2022-08-09
> > > > **Thank you for your engagement and helpful comments**
> > > >
> > > > Thank you so much for your engagement and helpful comments through this process -- our paper is now significantly clearer, thanks to your help. And yes, we do hope that this work can ignite a new chain of research going forward! We think that answering these questions is important both for understanding current models and for developing new models with in-context learning abilities.
> > > >
> > > > Best regards,
> > > >
> > > > The authors

---

### Official Review · Reviewer_v8kR · 2022-07-10

**Rating:** 7
**Confidence:** 3
**Soundness:** 3 good
**Presentation:** 3 good
**Contribution:** 4 excellent

**Summary:**

This paper asks a range of questions around what affects in-context vs in-weight learning in modern neural networks, focusing on Transformers, with RNN and LSTMs serving as comparisons. In totality, the experiments evaluate:

1. In-context vs In-weight based on burstiness of the data.
2. In-context vs In-weight based on number of training classes in the data.
3. In-context learning based on the number of labels per class.
4. In-context vs In-weight based on within-class variation.
5. How a Zipfian distribution in the data affects In-context and In-weight learning.
6. How In-context learning compares across Transformers, RNNs, and LSTMs.

The paper asks these questions in context of the recent strong LLM results. How can it be that these models seem to be exhibiting both in-context and in-weight learning? They then proceed to examine why this is and try to parse apart the causes by examining artificial data and comparing model classes.

**Questions:**

1. What is your actual definition of in-context vs in-weight learning?
2. How would you construct a language dataset to test some of these hypotheses instead of a dummy omniglot one? That feels like a missing experiment here because you do lots of variations, but do not include anything that gets at the environment that the LLMs are operating in. Could you create a language dataset that wasn't bursty? In my mind, this would be testing what happens when you use a real-world dataset that followed the uniform IID concept.
3. What would it take to do the same experiments in vision like you point to in the discussion?

**Limitations:**

As evidenced by the lengthy discussion, the authors understand the paper's limitations and what it points to.

**Strengths And Weaknesses:**

Originality:

This paper is original wrt the questions it asks. The methods to answering them (omniglot, simple plots, etc) are not new, but they don't need to be.

Clarity:

While the writing was good, the paper was surprisingly tougher to follow than I would have expected. I think it would get a huge improvement from a graphical interpretation of in-context vs in-weight learning. These are not terms I've seen before (and I've worked in FSL) and in-weight learning wasn't really ever properly defined in the paper. As far as I can tell, in-context learning was only roughly defined and that was on page 1 in relation to meta-learning. I had to piece together what I think is the definition from the rest of the work. I'm not sure if others you've shown this to have had the same problem, but just setting aside some space near the beginning to define what you mean by this would help a lot. Otherwise, I was left to wonder what actually fits into this view and what doesn't.

Quality:

The quality is high. Once I understood what was going on with their interpretation of in-context vs in-weight learning, the results were pretty interesting. I liked the Zipfian distribution experiment a lot because it was one of the questions I would have had towards doing this on language. I am genuinely surprised that the LSTM and RNN models failed so terribly and am curious if the authors have a take on this.

Significance:

The paper's experiments are intriguing and point to some insights into why LLMs are working so well. I'm not intimately familiar with this part of the literature to know whether these align with other results, but to me this is new and good stuff asking worthwhile questions with thought-provoking results. This strikes me as significant for computer vision as well because of the questions about whether it's worthwhile to shift to different paradigms of data than the prevailing uniform IID approach.

---

> ### Author Response · Authors · 2022-08-02
> **Thank you for your thoughtful review and questions (part 1)**
>
> Dear Reviewer,
>
> Thank you so much for your thoughtful review. We appreciate the effort you made to really understand the questions and concepts that we probe in this work. We have now rewritten the first paragraph of the introduction to make our definitions of in-context and in-weights learning much more explicit. We have also added bold font to the defined terms, to draw readers' attention and to make them easier to refer back to. Please let us know if any ambiguities remain:
>
> "Large transformer-based language models show an intriguing ability to perform in-context learning \citep{brown_language_2020}. This is the ability to generalize rapidly from a few examples of a new concept on which they have not been previously trained, without gradient updates to the model. In-context learning is a special case of few-shot learning in which the output is conditioned on examples from a 'context', and where there are no gradient updates. It contrasts with 'in-weights' learning, which is the standard setting for supervised learning – this is slow (requiring many examples), and depends on gradient updates to the weights."
>
> Thank you also for your questions, which we found very interesting. We answer below:
>
> > How would you construct a language dataset to test some of these hypotheses instead of a dummy omniglot one? That feels like a missing experiment here because you do lots of variations, but do not include anything that gets at the environment that the LLMs are operating in. Could you create a language dataset that wasn't bursty? In my mind, this would be testing what happens when you use a real-world dataset that followed the uniform IID concept.
>
> This is an interesting topic, and one we have thought carefully about. Our aim with these experiments was to factor out properties of data distributions that could promote the emergence of in-context learning, and also to test these properties in a domain outside of language – only by satisfying these two desiderata can we clearly show that it the data distributional properties themselves, rather than any other aspect of language, that drive in-context learning. This drove our decision to perform our experiments on the Omniglot image dataset, in addition to Omniglot being a standard dataset for few-shot or in-context learning. We would also argue that since the properties we investigated are well documented for natural language, they are in fact integral to the environment that LLMs are operating in.
>
> At the same time, our experiments do not provide causal evidence that these are precisely the properties driving in-context learning in LLMs, and we are careful to say only that our results hint at that interpretation. As you can imagine, given that these properties are inherent to natural language, we cannot remove these properties without breaking many other properties of language as well (syntax etc), and so that kind of ablation experiment on language cannot be done as far as we could tell. Nonetheless, we believe that it would be very interesting for future work to investigate the emergence of in-context learning in settings that replicate other aspects of language and language modeling, as mentioned in the discussion – e.g. having symbolic inputs, and training on next-token or masked-token prediction. We believe that another exciting future test of these ideas will be through the application of these ideas to larger-scale vision problems and reinforcement learning problems, as described in the answer to the next question.
>
> > What would it take to do the same experiments in vision like you point to in the discussion?
>
> Although we did perform our experiments on a small image dataset, it would be very interesting to apply the lessons of this work to applied vision work, or to larger vision datasets. We believe that there are two exciting potential routes worth investigating:
>
> (a) Adjust the distributions of vision datasets to reflect the properties found, in this work, to drive in-context learning.
>
> (b) Move to more naturalistic data, which will naturally reflect such properties – this seems to be a promising trend that is happening in the field in any case, and is reflected in the transition e.g. from ImageNet-style datasets a decade ago (with classes curated to be equal in size) to massive "in-the-wild" datasets like Ego4D, which are not curated to be artificially uniform in the same way. We look forward to seeing the results of training on these more natural datasets.
>
> (continued in part 2 below)

---

> > ### Author Response · Authors · 2022-08-02
> > **Thank you for your thoughtful review and questions (part 2)**
> >
> > > I am genuinely surprised that the LSTM and RNN models failed so terribly and am curious if the authors have a take on this.
> >
> > This is one of the directions we are most excited to pursue in future work. We were a bit surprised at these results as well, and hence performed the large hyperparameter search, and investigated both LSTMs and vanilla RNNs. But indeed, our results are in line with previous work, which has shown that recurrent architectures can only meta-learn to perform few-shot learning when the RNNs are complemented with additional architectural components, e.g. an additional memory component (Santoro et al, 2016) or an additional metric-learning component (Vinyals et al, 2016).
> >
> > We believe that recurrent models perform worse than transformers at few-shot learning potentially because of a combination of factors. Most importantly, perhaps, recurrent models lack the attention mechanisms that allow them to easily compare query items with items from context. Instead, they must pass information forward through the bottleneck of the hidden state, making it harder to bring forward all of the necessary information. They also suffer, as is well known, from exploding and vanishing gradients, which exacerbate the issue. We find this to be a very interesting question, and hope to investigate it in our continuing work.
> >
> > Thank you again for your thoughtful and careful review – we very much appreciate it.
> >
> > Best regards,
> >
> > The authors

---

> ### Author Response · Authors · 2022-08-08
> **Please let us know if you have any remaining questions**
>
> Dear reviewer,
>
> Thank you again for your review -- please let us know if you have any remaining questions, so that we can address them before the deadline tomorrow. Alternatively, if you feel that your original concerns about clarity are addressed, we would appreciate your updating your evaluation to reflect that.
>
> Thank you so much,
>
> The authors

---

> > ### Comment · Reviewer_v8kR · 2022-08-08
> > **Responding to update.**
> >
> > I don't have any further questions, and thanks for addressing one of my clarity concerns. I still see this as an Accept, well done.

---

> > > ### Author Response · Authors · 2022-08-08
> > > **Thank you**
> > >
> > > We appreciate your effort in critically evaluating our paper. Together with the other reviewers it has helped us to improve it. Thank you!

---

### Official Review · Reviewer_jRKE · 2022-07-11

**Rating:** 9
**Confidence:** 5
**Soundness:** 4 excellent
**Presentation:** 4 excellent
**Contribution:** 4 excellent

**Summary:**

This work performs a detailed investigation of the factors that promote different types of learning in transformers:  in-context vs. in-weights. The results delineate which aspects of the training distribution promote one form of learning or the other, or even both together. Further experiments find that recurrent models do not demonstrate the same learning abilities as the transformers studied here.

**Questions:**

How might the critical effects of the Zipfian exponent of the distributions be explained?

What are the color codings of the curves in Fig 7 & Fig 8? Are they just plotting the individual runs for different seeds, or perhaps for different hyperparameter settings?


**Limitations:**

Line 232 says:  “We performed a comprehensive hyperparameter search for all models (see Appendix for details).” But the appendix mentions only “15 runs for each architecture” over three hyperparameters with 2 values each. This seems like a fairly shallow hyperparameter sweep. Unless perhaps each hyperparameter combination was counted as a separate architecture, making 8 * 15 = 120 runs for the transformer alone, etc.

**Strengths And Weaknesses:**

This is an elegant and important paper. The investigations and their results are presented in a clear and compelling way, making for a real page-turner!

The work makes several contributions. The two types of learning studied here (in-context and in-weights) are neatly defined and explained. The experiments demonstrate how in-context learning is promoted by several specific properties of the training data:  burstiness, number of classes, and the Zipfian exponent of their marginal distributions. Especially exciting are the results indicating that a Zipfian marginal distribution with exponent of 1 over classes promotes a combination of both in-context and in-weights learning, which are otherwise at odds with each other in most experiments.

The comparisons between transformers and RNNs is of added benefit. The broader implications of these results, such as for application to RL, are especially exciting.

Overall, this work begins to dispel the mysteries behind the excellent empirical performance of transformers, and should lead to many more enlightening studies.

---

> ### Author Response · Authors · 2022-08-02
> **Thank you for your review and for the recognition**
>
> Dear reviewer,
>
> Thank you so much for your review and for your recognition of the value of this research – we have put in a lot of time and effort into this work, and we very much appreciate the acknowledgement.
>
> To address your questions:
>
> > How might the critical effects of the Zipfian exponent of the distributions be explained?
>
> This is a very interesting question – our hypothesis is that a Zipfian distribution has a combination of common and rare classes that each induces one type of learning. Common classes could be retained in weights because they were presented frequently enough to benefit from the kind of gradual learning that gradient descent performs. At the same time, the long tail of rare classes cannot be learned via gradient descent because they are each seen so infrequently – thus, the model must also acquire in-context learning abilities to handle these rare classes.
>
> In more detail, we believe that there may be two important factors: the dynamics of weight changes (which are determined by the optimizer, batch size, etc) and the Zipf exponent. Together, these factors establish the following:
>
> 1) The probability that a particular class is present in each subsequent batch. In the ideal supervised learning case, a particular class is observed in each batch, allowing its class label to be learned in the model's weights.
>
> 2) The amount that weights change from update to update, and how these changes contribute to catastrophic forgetting. For example, if it takes N batches before a particular class is seen again (see previous point), and the weights change by M, then it might be impossible for that class to be learned in the weights. What N and M are, we do not know.
>
> 3) The probability that a (catastrophically) forgotten class is present in each batch. In the ideal meta-learning case, each batch contains classes whose labels are forced to be unknowable (and hence, are functionally catastrophically forgotten).
>
> We predict that the probabilities of (1) and (3) need to be high to see the effects we observed. The particular conditions that make this so, conditioned on (2), are probably highly context dependent. We are very interested in exploring these questions and predictions in future work.
>
> > What are the color codings of the curves in Fig 7 & Fig 8? Are they just plotting the individual runs for different seeds, or perhaps for different hyperparameter settings?
>
> The colors are indeed indicating different runs, but they are not informative in any way beyond that. Thank you for pointing out this potential ambiguity – we have added clarification in the figure captions.
>
> > RE: Details of the hyperparameter search.
>
> The sweeps over max learning rate and num warmup steps were 15 log-uniform random samples over the ranges [1e-5, 0.1] and [1, 10000] respectively, rather than just two samples for each hyperparameter.  We have clarified this in the text.
>
> We performed 15 runs each for
> * the transformer with 2 layers
> * the transformer with 12 layers
> * the vanilla RNN with 2 layers
> * the vanilla RNN with 12 layers
> * the LSTM with 2 layers
> * the LSTM with 12 layers
> I.e. 90 runs total. We have clarified this in the text as well.
>
> Thank you again for your review and comments – they are much appreciated.
>
> Best regards,
>
> The authors

---

> > ### Comment · Reviewer_jRKE · 2022-08-05
> > **Reply**
> >
> > Thank you for the insights and clarifications!

---

### Official Review · Reviewer_D2K5 · 2022-07-16

**Rating:** 4
**Confidence:** 4
**Soundness:** 3 good
**Presentation:** 3 good
**Contribution:** 2 fair

**Summary:**

The authors pointed out that the reason why in-context learning (few-shot learning without parameter updates) can be achieved with large left-to-right (causal) language models is due to the data distribution.

Compared with in-weights learning (learning with parameter updates), the authors found that in-context learning is superior when the training data has the following characteristics: (1) burstiness (degree of unevenness as time series data), (2) large number of classes or heavy-tailed distribution (3) tokens have dynamic meaning. Also, they found that these characteristics are not valid in the recurrent architecture, but are more pronounced in the Transformer architecture.

**Questions:**

I assume "burstiness" refers to the so-called Taylor's law (bias as time series data), but can it be expressed in short training data of eight examples? Isn't it simply the uniformity of the frequency (cf. time series) that is changing...?

**Limitations:**

Limitations (future directions) are carefully described in Discussion section.

**Strengths And Weaknesses:**

### Strengths

It is fascinating to see research that provides an analysis from a new perspective on distributional properties of the training data to the important issue of realizing few-shot learning with causal language models.

### Weaknesses

The experiments are not controlled.

The three dependent variables are different in in-context learning and in-weights learning: (1) binary classification <-> multi-class classification (2) no model update <-> model update (3) classification for holdout classes <-> classification for trained (or common) classes. Therefore, we do not know which explanatory variable (burstiness, heavy-tail, dynamic semantics) works for which dependent variable. It must be said that the scientific findings are unclear.

---

> ### Author Response · Authors · 2022-08-02
> **Thank you for your review and comments (part 1)**
>
> Dear Reviewer,
>
> Thank you for your review. We were humbled to read that you found our research "fascinating", and that you appreciated the new perspective on an "important issue". Across the ML community, there has been widespread recognition that in-context learning is an important ability with many potential applications. However, this phenomenon is still poorly understood. In this work, we've provided clear evidence of factors that drive its emergence. We believe that the conclusions we reveal can be highly impactful – for understanding in-context learning in language models, and for potentially eliciting this useful ability in new domains.
>
> Thank you also for your questions regarding the dependent and explanatory variables – we have clarified below and in the paper, and we hope that these clarifications and additional experiments address your concerns.
>
> –
>
> Our experiments evaluate the effects of the distributional properties of training data on two primary dependent variables: performance on in-context learning, and performance on in-weights learning. Regarding the three variables you pointed to:
>
> (1) binary vs multi-class classification.
>
> Across training and both kinds of evaluation (in-context and in-weights), the model structure is the same – it is trained to predict one of a large number of labels (e.g. 1600 labels). For in-context evaluation, we reported accuracy based only on the two labels seen in context. This form of evaluation provides a more sensitive measure of the model’s in-context learning — for example, in full multi-class evaluation it is possible to achieve above-chance performance simply by randomly selecting from one of the labels in context. Also, the two-choice evaluation ensures that all experimental conditions have the same level of chance – otherwise, for the "number of training classes" and "dynamic meanings" experiments, chance level and difficulty would vary with the number of possible output labels.
>
> However, we do appreciate that this leads to an additional difference between the two evaluation conditions, as you point out. Therefore, we have now also computed and reported the in-context evaluations using full multi-class evaluation, while keeping in mind the caveat about chance levels changing with different output sizes. These results have been added in Appendix C.4. When computed in this way, the results do not change much. Given this full set of results on both two-choice and multi-class evaluation, we do not believe that this variable is a confounding factor.
>
> As a side note, the multi-class evaluation did uncover one interesting new result for the models trained on Zipfian distributions (we may pursue this in future work, and so are particularly glad that you brought up this question!). Feel free to read more in Appendix C.4 if you are interested.
>
> (2) no model updates vs model updates
>
> Learning image labels by relying on gradient updates (in-weights learning) and learning without model updates (in-context learning) are precisely our two dependent variables in this work. This factor is not a confound in our experiment – it is intrinsic to the phenomena we are attempting to measure! We have re-written the first paragraph of the introduction to make this much more explicit.
>
> (3) classification on holdout vs training classes.
>
> This is a good point. We made the decision to use a more standard form of evaluation by measuring in-context learning as the ability to learn an entirely held-out visual category (without model updates) . However, learning a new label binding for an otherwise familiar category is an equally valid form of in-context learning. We think the former is strictly more impressive than the latter, but to make things extra clear we have added results for the latter case (so that now, both in-weights learning and in-context learning can be measured based on 'familiar' classes from the training data). These results are reported in Appendex C.3. As you might predict, in-context learning emerges slightly more readily if defined in this way vs. the stricter definition we adopted originally. Otherwise, we observe very little difference in our empirical outcomes.
>
> (continued in part 2 below)

---

> > ### Author Response · Authors · 2022-08-02
> > **Thank you for your review and comments (part 2)**
> >
> >
> > > I assume "burstiness" refers to the so-called Taylor's law (bias as time series data), but can it be expressed in short training data of eight examples? Isn't it simply the uniformity of the frequency (cf. time series) that is changing...?
> >
> > As described in the introduction, burstiness is "a distribution that is not uniform across time, instead tending to appear in clusters". In statistics, computer science, and linguistics, there exist many proposed methods for quantifying burstiness, e.g. as described in the references in the introduction. A small sampling of examples include: a "numerical discrepancy" (the difference in counts between the current context and the mean), the Fano factor (a ratio between the variance and mean of counts), or the degree of skew when fitting an exponential distribution to the inter-event intervals. Taylor's law can indeed lead to one type of such burstiness, though it is only one instantiation of it. Rather than choosing a restrictive definition, we wanted to emphasize the general phenomenon, which is observed across many natural domains. We therefore used the simple and general definition stated in the introduction. For the experiments, we use the simple functional definition that bursty sequences have multiple occurrences of categories within a context window, chosen to reflect one type of burstiness that can be observed in e.g. language (within a single context window). We have added emphasis on this point in the Experimental Design section. Finally, it is indeed the case that burstiness implies nonstationarity (non-uniformity in frequency). We mentioned this in the introduction, though we call it "a distribution that is not uniform across time" (let us know if any alternate wording might be preferred).
> >
> > We hope that we were able to address your questions through these explanations, additional analyses, and updates to the paper – please let us know if you have any remaining questions at all.
> >
> > Best regards,
> >
> > The authors

---

> ### Author Response · Authors · 2022-08-08
> **Please let us know your thoughts on our rebuttal**
>
> Dear reviewer,
>
> Thank you again for your review -- please let us know if you have any remaining questions or concerns, so that we can address them before the deadline tomorrow. Alternatively, if you feel that your original concerns are addressed, we would appreciate your updating your evaluation to reflect that.
>
> Thank you so much,
>
> The authors

---

### Meta-Review · Area_Chair_d23H · 2022-08-26

**Recommendation:** Accept
**Confidence:** Certain

**Metareview:**

This paper poses and analyses an interesting question -- do statistical properties of the training data affect emergent behavior (e.g. in-context learning) in Transformers? The study is novel since it probes the properties of the data itself, as opposed to most existing work that has studied the effect of model architectures and training algorithms on model capabilities. The paper shows that properties like Zipfian distribution and burstiness are useful for eliciting in-context learning (two properties that are widely found in natural languages, where in-context learning has had a rapid emergence). All the reviewers appreciated the originality and timeliness of this paper, and experiments provide interesting conclusions. Overall, I believe the paper will spark future inquiry into data distributional properties.

I recommend the authors pay attention to Reviewer D2K5's comment about clarifying the definition of "burstiness" in the writing and making it more consistent with the experiments performed.

**Award:**

No

---

### Decision · Program_Chairs · 2022-09-14

Accept